# Based on Transcriptome Sequencing of Cell Wall Deficient Strain, Research on Arabinosyltransferase Inhibition’s Effect on the Synthesis of Cell Wall in *Chlamydomonas reinhardtii*

**DOI:** 10.3390/ijms242417595

**Published:** 2023-12-18

**Authors:** Wenhua Zhang, Menghui Shang, Lexin Qiu, Bin Liu, Xiaonan Zang

**Affiliations:** 1Key Laboratory of Marine Genetics and Breeding, Ministry of Education, Ocean University of China, Qingdao 266003, China; zhangwenhua1358@163.com (W.Z.); shangmenghui11@163.com (M.S.); lexinq1998@163.com (L.Q.); 2Yellow Sea Fisheries Research Institute, Qingdao 266003, China

**Keywords:** *Chlamydomonas reinhardtii*, transcriptome sequencing, arabinosyltransferase, cell wall glycoprotein, gene knockout

## Abstract

To explore the key genes involved in cell wall synthesis and understand the molecular mechanism of cell wall assembly in the model alga-*Chlamydomonas reinhardtii*, transcriptome sequencing was used to discover the differentially expressed genes in the cell wall defective strain. In the glucose metabolism, lipid metabolism, and amino acid metabolism pathways, the gene expressions involved in the synthesis of cell wall functional components were analyzed. The results showed that in the cell wall defective strain, arabinosyltransferase gene (*XEG113*, *RRA*) related to synthesis of plant extensin and some cell wall structural protein genes (*hyp*, *PHC19*, *PHC15*, *PHC4*, *PHC3*) were up-regulated, 1,3-β-glucan synthase gene (*Gls2*) and endoglucanase gene (*EG2*) about synthesis and degradation of glycoskeleton were both mainly up-regulated. Then, ethambutol dihydrochloride, an arabinosyltransferase inhibitor, was found to affect the permeability of the cell wall of the normal strain, while the cell wall deficient strain was not affected. To further research the function of arabinosyltransferase, the *RRA* gene was inactivated by knockout in the normal cell wall algal strain. Through a combination of microscope observation and physiological index detection, it was found that the cell wall of the mutant strains showed reduced structure levels, suggesting that the structure and function of the cell wall glycoprotein were weakened. Therefore, arabinosyltransferase may affect the glycosylation modification of cell wall glycoprotein, further affecting the structure assembly of cell wall glycoprotein.

## 1. Introduction

The cell wall is widely present in non-animal species and plays an important role in the maintenance of life. In eukaryotic algae and plants, there are significant similarities in the structure and composition of cell walls. Deeply understanding the synthesis mechanism of cell walls is of great significance for elucidating the basic laws of cell activity and the evolution process from algae to plants. Extensin was initially discovered in plant cell walls, and more and more studies have shown that hydroxyproline-rich glycoproteins (HRGPs) play similar roles in eukaryotic algal cells, but the function and synthesis mechanism of HRGPs in the synthesis of algal cell walls are still unclear. Therefore, the model microalgae *Chlamydomonas reinhardtii* with a clear genetic background was used as the research object to study the mechanisms related to HRGP synthesis, laying the foundation for elucidating the role of extensin in cell wall synthesis and the cell evolution of algae towards plants.

*C. reinhardtii* is a unicellular eukaryotic green alga belonging to Chlorophyta, Volvocales. It is an important link in the evolution of multicellular algae and higher plants. *C. reinhardtii* also has the characteristics of protists, so early scholars mainly used it as the study object of cell structure [1]. After that, various mutant types of *C. reinhardtii* were obtained by physical and chemical methods, among which the discovery of cell wall deficient strains laid the foundation for the study of cell wall structure and provided favorable conditions for the successful establishment of an efficient genetic transformation system in *C. reinhardtii*. This history makes *C. reinhardtii* one of the important model organisms in biological research [2], and it is widely used in microalgae biotechnology [3].

In the early days, the cell wall composition of *Chlamydomonas* is considered to be cellulose [4]; however, a growing number of studies have shown that the cell wall of *C. reinhardtii* is a kind of glycoprotein lattice structure rich in hydroxyproline, similar to the plant extensin of higher plants. The analysis of amino acid composition showed that the cell wall protein was rich in hydroxyproline residues, and the chromatographic analysis showed that the glycogroup components in the cell wall were mainly arabinose, galactose, and a small amount of mannose [5,6,7,8,9,10]. In addition, when the defective cell wall strains of *C. reinhardtii* were taken as the research object, it was found that the glycoprotein layer of the cell wall may have a self-assembly process [11,12,13,14]. Recently, Cronmiller et al. (2019) have carried out functional studies on some cell wall genes, including some cell wall protein processing genes and structural protein genes. Concurrently, it was found that these cell wall genes were partially activated after lysin treatment and participated in the cell wall integrity signal regulation in *C. reinhardtii* [15]. In addition, it has been shown that the arabinosyltransferase encoded by *RRA* was involved in protein O-glycosylation during root hair development of model organism *Arabidopsis thaliana*, and played an important role in stabilizing the helical conformation of cell wall glycoprotein [16]. Importantly, homologous genes belonging to the glycosyltransferase family (GT77) were also found in *C. reinhardtii*, which confirmed that the cell wall glycoproteins of green algae shared a common ancestor with the extensin of higher plants in the origin and evolution of green algal cell walls [17].

RNA sequencing is a technology that comprehensively studies gene function and reveals molecular mechanisms in specific biological processes [18]. The transcriptome is a collection of RNAs transcribed by a specific tissue or cell at a certain development stage or functional state. By studying the transcriptome, the gene expression level can be understood at the overall level, and the molecular mechanism of gene expression regulation can be revealed [19]. At present, the molecular studies on the functional components in *C. reinhardtii* cell wall are in the development stage. The functions of some cell wall genes have been studied, and the self-assembly process in *C. reinhardtii* cell wall has been confirmed. However, the key genes and their roles in the cell wall synthesis process are still unclear, and more studies are needed at the molecular level to truly understand the assembly mechanism of *C. reinhardtii* cell wall. In the Chlamydomonas Resource Center (University of Minnesota), there were many mutant cultures of *C. reinhardtii.* Among them, CC-849 is a cell wall deficient strain with no other significant differences compared to the wild type, which lays a good foundation for researching the differences between cell wall synthesis and assembly. Therefore, this work aims to explore the key genes involved in cell wall assembly in the cell wall defective strain of *C. reinhardtii* by transcriptome sequencing technology and then research the function of arabinosyltransferase on the formation of cell wall glycoprotein through gene knockout, which lays a foundation for further reveal the role of extensin in cell wall synthesis and the evolution of *C. reinhardtii* cell wall glycoprotein and higher plants extensin.

## 2. Results

### 2.1. Structural Analysis of the Cell Wall of the Two Strains of C. reinhardtii

Both stains of CC-849 and CC-124 were in a logarithmic phase at about 1–3 days and reached a stable phase at about 5 days; they were then used for structural analysis of the cell wall (Figure 1a). Through transmission electron microscopy shown in Figure 1b, it was found that the algal cells of CC-849, a cell wall deficient strain, were mostly irregular in shape, and the absence of its cell wall resulted in little glycoprotein layer structure. Concurrently, the algal cells of CC-124 with normal walls were mostly round or oval, with complete cell wall structure and obvious structural layers could be observed. From the outer plasma membrane of the cell, cell wall layers are defined as W1-W7 successively, among which the central glycoprotein lattice structure is divided into three layers, including the inner dense layer (W2), the central granular layer (W4) and the outer crystalline layer (W6), which is consistent with the previous description by Goodenough et al. [9].

In addition, the sensitivity of the two kinds of algal cells to the lysing agent NP-40 was further detected to compare cell wall permeability, which was positively correlated with the ratio of F_2_/F_1_. The two algal strains were set with the same initial cell density, cultured under the same conditions, and sampled every day to detect the sensitivity to NP-40. The growth trend of the two algal strains in a week with logarithmic and stable phases. The results showed that CC-849 with the defective cell wall was more easily erupted and released more chlorophyll, which resulted in a higher fluorescence ratio of F_2_/F_1_ detected after and before treatment. As a consequence, CC-849 was more sensitive to NP-40 than CC-124 during the growth period, especially on the third day, with greater cell wall permeability, as shown in Figure 2.

### 2.2. Transcriptome Sequencing Data Quality Control and Sequence Alignment

To investigate the differential expression genes of CC-849 and CC-124 in cell wall synthesis, the two algal strains in the stable stage were sampled and sequenced transcriptome. The cell wall deficient strain CC-849 was the experimental group, denoted as CW-D; normal cell wall strain CC-124 was the control, denoted as CW, and each group consisted of triplicate. All the analyses were based on clean, high-quality data. After sequencing quality control, a total of 46.29 Gb of clean data was obtained, and the percentage of Q30 bases in each sample was no less than 95.10%. Only reads with a perfect match or one mismatch were further analyzed and annotated based on the reference genome. According to the comparison results, the efficiency of comparison between reads of each sample and reference genome ranged from 95.78% to 96.42%, and the efficiency that reads of each sample were uniquely mapped to the reference genome ranged from 91.78% to 92.80%, as shown in Table 1.

### 2.3. Analysis of Differentially Expressed Genes

#### 2.3.1. Summary of DEGs

To explore the difference in gene expression in cell wall synthesis between algae, the following comparison group was CW−D1&CW−-D2&CW−D3 vs. CW1&CW2&CW3, and Fold Change ≥ 2 and FDR < 0.01 were used as the criteria for differential gene screening. According to the statistics of results, there were 4259 differentially expressed genes, among which the number of up−regulated genes (2365) was significantly more than the number of down−regulated genes (1894). Similarly, when making a clustered heat map for this comparison, as shown in Figure 3, it also indicates that there are more gene expressions in the cell wall deficient strain CC-849 and more genes annotated to the metabolism pathway in the KEGG database. To study the functional information of DEGs, it is annotated in eight databases, as shown in Table 2.

#### 2.3.2. GO Enrichment Analysis of DEGs

The total differential genes were classified by GO enrichment (corrected *p*-value < 0.05), and the results showed that the differential genes were significantly enriched in the “translation” of biological processes, “chloroplast” of cell components, and “ATP binding” of molecular function. The up−regulated differential genes were significantly enriched in “rRNA processing”, “nucleus”, and “ATP binding”. Down−regulated differential genes were significantly enriched into the “translation”, “integral of component membrane”, and “structural constituent of ribosome”. In general, differential genes were mainly involved in the translation process and may be related to the molecular activity of cellular structures.

#### 2.3.3. KEGG Enrichment Analysis of DEGs

The total differential genes in KEGG enrichment were significantly enriched in 20 metabolic pathways, including “ribosome”, “biosynthesis of amino acids”, and “ribosome biogenesis in eukaryotes”. The up−regulated differential genes were significantly enriched in “ribosome biogenesis in eukaryotes”, “biosynthesis of amino acids”, and “DNA replication” pathways. Down−regulated differential genes were significantly enriched in the “ribosome”, “valine, leucine, and isoleucine degradation”, and “porphyrin and chlorophyll metabolism” pathways. In general, in the comparison group, the expression of differential genes was mainly related to ribosome, amino acid synthesis and degradation, and carbon metabolism pathways.

### 2.4. Metabolic Pathways Related to Cell Wall Components

#### 2.4.1. Synthetic Pathway of Plant Extensin

As for *C. reinhardtii* cell wall, the glycoprotein component cross-linked to the microfiber matrix is the main structure, rich in hydroxyproline, which is similar to the extensin of higher plants. In the synthetic process of extensin in higher plants, proline hydroxylation is the first step, and Hyp and Ser residues are further modified by O−glycosylation after the synthesis of hydroxyproline, including arabinylation of Hyp and galactosylation of Ser [20]. In the metabolic pathway found in the comparison group of CC-849 vs. CC-124 shown in Figure 4, arabinosyltransferase genes (*XEG113*, *RRA*) were up-regulated, and peptidyserine α−galactosyltransferase gene (*SGT1*) was down-regulated. These results indicate that there was an active post-translational modification process of glycoprotein in CC-849. Perhaps it is the lack of cell wall in CC-849 that leads to the compensatory expression of cell wall components such as extensin.

#### 2.4.2. Cell Wall Structural Proteins

Recent research noted that cell wall proteins of *C. reinhardtii* contained protein-processing types, such as glycosyltransferase described above, and structural protein types, which played roles in hydroxyproline-rich protein and protein pherophorin, respectively [15]. In addition to transcriptional differences among glycosyltransferase genes at the metabolic level, cell wall structural protein genes (*hyp*, *PHC19*, *PHC15*, *PHC4*, *PHC3*) showed significant up-regulation (Table 3), verifying the existence of active cell wall genes expression process in CC-849. It was further speculated that algal cells lacking normal cell walls would constantly attempt to rebuild the cell wall, drive transcriptional activation of cell wall genes, and induce local cell wall synthesis. Accordingly, these findings are consistent with the fact that the absence of the cell wall itself is a signal to drive the transcriptional activation of cell wall protein processing genes and cell wall structural protein genes, which may jointly participate in the self-assembly process of the cell wall glycoprotein layers.

#### 2.4.3. Synthesis and Degradation Pathways of Glycoskeleton

In addition to glycoprotein structure, the matrix of the glycogroup network is also an important structure of the cell wall in *C. reinhardtii.* In the metabolic pathway of cellulose synthesis and degradation, the metabolic involved in the glycoskeleton was analyzed (Figure 5). Although cellulose synthase (β-1, 4-glucan synthase) is the crucial enzyme in the synthesis of cellulose in some plant cell walls [21], transcriptome results manifested no up-regulation and down-regulation difference shown as CesA in the figure. Some studies have shown that glucose monomers are linked via β-1, 3-glycosidic bonds in bacteria and algae [22]. The 1, 3-β-glucan synthase gene *Gls1* in the synthetic pathway was down-regulated, and *Gls2* was up-regulated in CC-849. The endoglucanase gene *EG1* in the degradation pathway was down-regulated, and *EG2* was up-regulated. Additionally, the difference in up-regulated genes was more significant than that in down-regulated genes, indicating that the expression level of genes in the degradation process may be stronger than that in the synthesis process, and the synthesis and degradation processes of glycoskeleton in CC-849 were more active, which was consistent with the enhanced protein synthesis in CC-849.

### 2.5. Real-Time PCR Verification

Targeting metabolic pathways associated with the synthesis of cell wall components, *XEG113*, *RRA,* and *SGT1* in the synthesis of plant extensin, five cell wall structural protein genes *hyp*, *PHC19*, *PHC15*, *PHC4* and *PHC3*, four genes *Gls1*, *Gls2*, *EG1*, *EG2* in the synthesis and degradation of glycoskeleton were selected for quantitative validation of gene expression using qRT-PCR. Real-time fluorescence quantitative PCR verification (Figure 6) was consistent with the transcriptome sequencing results, proving that the transcriptome results were effective.

According to the results of transcriptome sequencing and quantitative PCR validation of gene expression, the difference in *RRA* between the two algal strains was significant. Additionally, *RRA* gene coding glycosyltransferase is one of the most crucial genes related to the synthesis of plant extensin. Thus, the function of glycosyltransferase and its coding gene *RRA* was further studied.

### 2.6. Effects of the Arabinosyltransferase Inhibitor on the Structure of Cell Wall

Ethambutol dihydrochloride is a dihydrochloride that can inhibit the activity of arabinosyltransferase [23]. By comparing the cell morphology of the two algal strains treated with different concentrations of ethambutol dihydrochloride and further testing the sensitivity to NP-40, as shown in Figure 7, it was found that different concentrations of ethambutol dihydrochloride had little effect on the growth state and the permeability of cell wall of CC-849. Instead, the growth state of CC-124 was affected significantly at 30, 40, and 50 mg/L of inhibitor, and the permeability of the cell wall was all increased after the addition of the inhibitor, of which the greatest increase in the permeability of the cell wall occurred at 50 mg/L inhibitor concentration. Similarly, the cell morphology of CC-124 treated with 50 mg/L of inhibitor showed a transparent algal cell structure, which indicated the cell morphology and structure of CC-124 were greatly affected by 50 mg/L inhibitor.

This suggests that glycosyltransferase may play an important role in cell wall synthesis in *C. reinhardtii*. Next, the *RRA* gene coding glycosyltransferase tended to be knocked out to study its function further.

### 2.7. Cell Wall Analysis of Mutant Strains with Arabinosyltransferase Gene Knockout

#### 2.7.1. Information on Arabinosyltransferase Gene

According to the transcriptome data, the gene sequence of arabinosyltransferase gene (*RRA*) is significantly differentially expressed in the synthesis and assembly of cell wall glycoprotein, and the gene sequence of *RRA* in *C. reinhardtii* was successfully cloned. The length of *RRA* is 1017 bp without introns, encoding 368 amino acids, and the theoretical molecular weight is 41.78 kDa. The amino acid sequence of *RRA* has a conserved domain of nucleotide-diphospho-glycotransferase, which belongs to the glycosyltransferase superfamily with a short DxD conserved motif. Concurrently, the glycosyltransferase family was investigated, and homologous genes with the same conserved domain were found in other algal strains and higher plants by Blastp analysis of the *RRA* coding protein sequence of *C. reinhardtii* (Figure 8), which revealed the evolution trend of HRGPs from green algae cell wall to plant cell wall extensin.

#### 2.7.2. Screening of Mutants

To study the function of arabinosyltransferase gene, transgenic strains of *C. reinhardtii* were created. The constructed *RRA* gene knockout vector (Figure 9a) was introduced into CC-124. Using the upstream and downstream sequences of the *RRA* gene as homologous recombination sites, the antibiotic resistance gene (*hyg*) replaced most of the *RRA* gene. The positive colonies were screened from the plate containing hygromycin, and PCR was performed with the primers of *RRA* and *hyg* genes to verify the positivity of the selected colonies. As shown in Figure 9b, by PCR performing with primers of *RRA*, all mutants have a band at about 2–3 kb on 1% agarose gel electrophoresis, which was identical to the corresponding sequence in the knockout vector but different from the amplified band of CC-124. Further, the 969 bp sequence, including part *RRA* and part *hyg* gene, was gene sequenced using RRA-F and hyg-R as primers, which were all consistent with the vector sequences, as shown in Figure 9c. Therefore, these selected strains were identified as positive and named 124-SR1-10, respectively.

A real-time fluorescence quantification technique was utilized to further detect the expression of the *RRA* gene at the transcriptional level of transformed algal strains. The negative logRQ value showed that the transcription level of the *RRA* gene was lower in each knocked-out algal strain than that in the control CC-124, with an average decrease ratio of 3.662 (Figure 9d). This suggests that the transcription of the *RRA* gene was inhibited severely in the knocked-out algal strains.

#### 2.7.3. Comparative Analysis of Cell Wall among Mutants

The positive mutant strains 124−SR1−10 were selected to measure their growth and sensitivity to NP-40. It was found that the sensitivity to NP-40 of each mutant was higher than that of the control CC-124, indicating that the cell wall permeability of each mutant increased to varying degrees. Consequently, the growth density of mutants decreased significantly in contrast to the control CC-124 (Figure 10a). As shown in Figure 10b, the mutant 124-SR2 with obvious cell wall defection and the mutant 124-SR8 with relatively good growth and certain cell wall defection would be further selected for microscope observation.

After incubation with FITC-RCA I, the surface of wild-type algal cells emitted strong fluorescence because there were lots of glycogroup that could bind FITC-RCA I. Although the emitting fluorescence of the mutant algal cells almost cannot be observed, which indicates that little target glycogroup existed in the cell wall or the glycoskeleton structure had changed, thus almost cannot be labeled by FITC-RCA I (Figure 10c). The change in fluorescence intensity of FITC-RCA I in the single algal cell between the control strain and the mutants demonstrated that the mutants had a weak ability to bind FITC-RCA I (Figure 10d). Both of the mutants showed reduced fluorescence intensity compared to CC-124, and 124-SR2 was even lower than 124-SR8, which meant the glycoskeleton structure of the cell wall of the mutants 124-SR2 and 124-SR8 was changed.

The mutants 124-SR2 and 124-SR8 were further analyzed for electron microscope observation (Figure 10e). It was found that the algal cells of 124-SR2 became clumped together, the cell morphology became irregular, the cell wall became thinner, and the structural levels of the cell wall became unclear and decreased. Additionally, the cell wall of 124-SR8 also became unclear, and the structure was loose. We further discovered, through measurement, that the thickness of the cell walls of 124-SR2 and 124-SR8 reduced significantly compared to CC-124 (Table 4).

## 3. Discussion

*C. reinhardtii* has the advantages of simple culture conditions, a short growth cycle, and a clear genetic background, which makes it a good material for many transgenic technology studies. The successful screening of cell-deficient strains created more favorable conditions for the genetic transformation of *C. reinhardtii* and laid a solid foundation for the study of cell structure at the same time. The cell wall of *C. reinhardtii* is dominated by glycoproteins rich in hydroxyproline. During its formation, the fiber network of the inner layer of W1 and the outer layer of W7 is first synthesized. After the initial complex is constructed, other cell wall layers are gradually assembled in it [11]. The hydroxyprolin-rich glycoprotein (HRGPs) in the inner layer is initially synthesized in the endoplasmic reticulum and Golgi apparatus, transported with vesicles to the shrinking vacuole, and released to the cell surface for self-assembly [12]. Finally, the glycoproteins interact and cross-link to the inner and outer fiber layers. The highly organized HRGPs in algal cell walls are ancient and primitive and have evolved into a superfamily containing multiple groups in plant cell wall extensin [24]. In the study concerning the origin and evolution of cell walls in green algae, it has been stated that the extensin of higher plants is a cell wall protein that shares a common ancestor with the HRGPs of the cell wall of green algae [17]. From the hydroxyproline-rich glycoproteins found in the algal cell wall to the development of higher plant extensin, it is the secular evolution of plant cell wall structure and function.

In this study, based on the comparative analysis of the cell wall component synthesis between the cell wall defective strain and normal cell wall strain, the differentially expressed genes were mainly related to the extensin synthesis pathway and synthesis and degradation pathways of glycoskeleton in *C. reinhardtii*. In CC-849, there were up-regulated *XEG113* and *RRA* (arabinosyltransferase gene) and downregulated *SGT1* (peptidyserine α-galactosyltransferase gene) in the process of plant extensin synthesis. Additionally, some cell wall structural protein genes were up-regulated. So, the expression of various regulatory genes was speculated to cooperate to maintain the level of cell wall component metabolism. However, these genes are generally up-regulated, indicating active cell wall metabolism, which seems to be inconsistent with the defect of the cell wall in CC-849. It is speculated that the defect of the CC-849 cell wall may not be entirely caused by the expression of these genes, which are up-regulated to synthesize missing cell walls to compensate for the CC-849 cell wall defect. A previous study demonstrated that gamete-specific (*GAS*) Hyp-rich pherophorin-encoding genes (*GAS28*, *GAS30*, *GAS31*), cell wall pherophorin gene (*PHC19*) and protein processing genes (*SEC61G*, *AraGT1*, *RHM1*) related to translation or glycosylation indicated a substantial increase in gene expression upon g-lysin treatment, which in the same way suggested that cell wall removal up-regulated cell wall genes expression [15].

Hydroxyproline-rich glycoproteins in higher plant cells, including extensin (EXTs), are secreted into the cell walls after synthesis and shaped by posttranslational modifications. There are O-glycosylation chains with up to 4 or 5 linear arabinosyl units on each Hyp and monogalactosylation of Ser residues in Ser (Hyp)**_4_** repeats of EXTs (Figure 11). Typically, during this process, there are three groups of arabinosyltransferases genes (*AraTs*), hydroxyproline O-arabinosyltransferases genes (*HPAT1-HAPT3*; GT8 family), reduced residual arabinosyltransferase genes (*RRA1-RRA3*) and xyloglucanase gene (*XEG113*; GT77 family), and a peptidyl-Ser galactosyltransferase gene (*SERGT1*; GT96 family) adding a single galactopyranose to each Ser residue in Ser (Hyp)**_4_** motifs. Ultimately, the glycosylated EXTs may be cross-linked with the peroxidases at the Tyr residue to construct a three-dimensional network that may interact with other components of the cell walls. Some glycosylation modifying enzymes have been shown to play a role in root hair development in the model organism *Arabidopsis thaliana* and influence self-assembly of the cell wall EXTs [25]. Studies have found that *Chlamydomonas* also had multiple proline hydroxylase genes (*P4H*) that were essential for the correct assembly of the cell wall. By knocking out the *P4H* gene in *Chlamydomonas*, the cell wall layers were also unclear, and it was speculated that the disappearance of W2 and W4 layers resulted in the merger of W1 and W7 layers. It was speculated that *P4H* might affect the hydroxylation of proline, reduce the content of Hyp, and further affect its glycosylation, thus affecting the cross-linking and stability of the fiber layer [26]. However, the function of some O-glycosylation modifying enzymes is rarely reported in the research of algae.

In the present study, as *RRA* gene coding glycosyltransferase is one of the most crucial genes related to the synthesis of plant extensin found by transcriptome sequencing, arabinosyltransferase inhibitor was found to prevent the algal cell growth and increase cell permeability. At the same time, after arabinosyltransferase gene (*RRA*) was knocked out, the cell wall showed blurred boundaries and reduced structural levels. The structure of W2 and W6 layers in the cell walls of mutant strains did not seem to disappear, but the boundary is not clear. Additionally, the reason for the looseness of the cell wall structure may be that the W7 layer is not cross-linked, and neither is the W1 layer. Similar to the CC-849 cell wall defective type, there was a vague structure layer on the outside of the plasma membrane, which may resemble the remnants of the W1 and W7 layers. Therefore, arabinosyltransferase may play an important role in the glycosylation modification and conformational stability of cell wall glycoprotein, which further affects the structural assembly of cell wall glycoprotein in *C. reinhardtii.* Similarly, during the root hair development of *A. thaliana*, the knockout of arabinosyltransferase genes (*RRA1* and *RRA2*) revealed reduced arabinose content in the cell wall, demonstrating that these genes involved in the process of protein O-glycosylation and played an important role in stabilizing the helical conformation of cell wall extensin [27]. At the same time, the glycosyltransferase family was investigated, and homologous genes were found in *C. reinhardtii* with the same conserved domain, which confirmed the evolution trend of HRGPs from green algae cell wall to plant cell wall extensin.

## 4. Materials and Methods

### 4.1. Cultivation and Treatment of Algae

*C. reinhardtii* was obtained from the Chlamydomonas Resource Center (University of Minnesota laboratory), and the algal species number of the cell wall deficient strain is CC-849, and the wild-type strain with normal cell wall is CC-124. The cells of the two algal strains were cultured in a Tris-acetoacetate-phosphate (TAP) medium (pH 6.8). The illumination was 50 μmol·m^−2^ s^−1^, the temperature was 23 ± 1 °C, the light period was 12L:12D, and the rotating speed was 100 rpm. The two algal strains were set with the same initial cell density and cultured under the same conditions for three days, and the algal strains in the stable stage were sampled. After freezing with liquid nitrogen, the samples were stored at a low temperature of −80 °C for RNA extraction. The experimental groups were as follows: cell wall deficient strain CC-849 was the experimental group, denoted as CW-D, normal cell wall strain CC-124 was the control, denoted as CW, and each group consisted of triplicate.

### 4.2. Permeability and Integrity Analysis of Cell Wall

The biomass of *C. reinhardtii* was characterized by the number of cells per milliliter of algal culture and calculated from the absorbance OD_750_. According to the experimental method of Yali Wang (2016) [28], the lysing agent NP-40 (Solarbio, Beijing, China) with the main composition of Tris (pH7.4), NaCl, and 1% NP-40 was used to compare cell wall permeability of the two algal cells. The fluorescence emission value F_1_ was measured at 680 nm under an excitation light of 435 nm by taking a one-milliliter suspension from the algal culture of CC-849 and CC-124, which were in stable stages. After treatment with NP-40 for 1 h and centrifugation at 3000× *g* for 5 min, the fluorescence value F_2_ was measured by taking a one-milliliter supernatant at the same set wavelength. The ratio of F_2_/F_1_ was positively correlated with the cell wall permeability.

In addition to the measurement of physiological parameters, the integrity of the *C. reinhardtii* cell wall was analyzed from the following two aspects. Firstly, fluorescein-labeled ricinus communis agglutinin I (FITC-RCA I, Qiyue, Xian, China) was used to show the location of the glycogroup on the surface of algal cells, which can bind each other specifically and directly. In reference to the method of Juan Lin (2020) [29], the algal culture at the same period and concentration was incubated with FITC-RCA I in the dark for 1 h, rinsed three times with PBS after centrifugation, and then green fluorescence was observed under a confocal laser scanning microscope (CLSM). Apart from observing the glycogroup structure of the algal cell walls, the overall structure level of the algal cell walls was further observed using transmission electron microscopy (TEM).

### 4.3. Extraction of Total RNA and Construction of cDNA Library

RNA was extracted from CC-849 and CC-124 by using an RNA extraction kit (Biomarker, Beijing, China). RNA concentration and purity were measured using Nanodrop 2000 and RNA integrity was assessed using the RNA Nano 6000 Assay Kit of the Agilent Bioanalyzer 2100 system to ensure transcriptome sequencing with qualified RNA samples. After the samples were qualified, transcriptome sequencing libraries were generated using the NEBNext UltraTM RNA Library Prep Kit for Illumina (NEB, Ipswich, MA, USA) following the manufacturer’s recommendations, and index codes were added to attribute sequences to each sample.

### 4.4. Sequencing, Quality Control and Comparative Analysis

After the library was constructed, the effective concentration of the library was accurately quantified by qRT-PCR (the effective concentration of the library was higher than 2 nM) to ensure the quality of the library. After the database check was qualified, the clustering of the index-coded samples was performed on a cBot Cluster Generation System using TruSeq PE Cluster Kit v4-cBot-HS (Illumina, San Diego, CA, USA). After cluster generation, the library preparations were sequenced on an Illumina platform, and paired-end reads were generated.

The raw reads were further processed with a bioinformatic pipeline tool, BMKCloud (www.biocloud.net accessed on 15 December 2023) online platform. Raw data (raw reads) of FastQ format were first processed through in-house Perl scripts. In this step, clean data (clean reads) were obtained by removing reads containing adapter, reads containing ploy-N, and low-quality reads from raw data. At the same time, Q20, Q30, GC-content, and sequence duplication levels of the clean data were calculated. All the downstream analyses were based on clean data with high quality. These clean reads were then mapped to the reference genome sequences using the Hisat2 tools. Based on the reference genome of *C. reinhardtii* (https://www.ncbi.nlm.nih.gov/genome/147 accessed on 15 December 2023), only reads with a perfect match or one mismatch were further analyzed and annotated.

### 4.5. Differentially Expressed Gene Analysis

The FPKM value (number of fragments per kilobase length per million fragments from a gene) was used to determine the expression pattern of differentially expressed genes under different experimental conditions [30]. For samples with biological duplication, DESeq2 was suitable for differential expression analysis between sample groups to obtain differential expression gene sets between two biological conditions. In the process of differential expression gene detection, Fold Change ≥ 2 and error discovery rate (FDR) < 0.01 were used as screening criteria, both of which were obtained by correcting the *p*-value of difference significance. To investigate the differential expression genes of CC-849 and CC-124 in cell wall synthesis, a comparative combination of CW-D1&CW-D2&CW-D3 vs. CW1&CW2&CW3 was set.

The above-obtained genes were functionally annotated in seven databases (Nr, Nt, Pfam, KOG/COG, Swiss-prot, KEGG, and GO). According to the functional annotation results, the classification methods of different databases can be classified. Through GO annotation, gene functions were classified into three categories (biological processes, cell components, and molecular functions). After KO annotation, gene annotations were classified according to the KEGG metabolic pathway in which they participate.

### 4.6. Quantitative PCR Validation of Gene Expression

Based on the transcriptome results, differentially expressed genes related to the synthesis of cell wall components in glucose metabolism, lipid metabolism, and amino acid metabolism were selected for quantitative validation of gene expression using qRT-PCR. *β-tubulin* gene was selected as the internal reference gene.

RNA extraction was conducted using the E.Z.N.A.^®^ Plant RNA Kit (OMEGA, Germany), and the integrity of the 28S and 18S bands were checked by agarose gel electrophoresis to verify the RNA quality. The total RNA was reverse transcribed into cDNA, and the nucleic acid concentration was determined by Nanodrop. qPCR was performed using the TB Green Premix Ex Taq II Kit (Tli RNaseH Plus, Takara, Dalian, China), and each sample was set up in triplicate. The experimental data were analyzed and plotted according to the RQ = 2^−ΔΔCt^. qPCR genes and primers are shown in Table 5.

### 4.7. Treatment with Arabinosyltransferase Inhibitor

According to the differential expression genes obtained through transcriptome sequencing (4.5) and their validation experiments (4.6), the difference in arabinosyltransferase gene (*RRA*) between the two algal strains was significant. To research its role in the synthesis of cell walls in *C. reinhardtii*, the inhibitor-ethambutol dihydrochloride (Sparkjade, Shandong, China) is used to inhibit the activity of arabinosyltransferase.

According to pre-experimental results, the inhibitor began to affect the growth of normal cell wall strain CC-124 at a concentration of 20 mg/L. Thus, the inhibitor concentration gradient was set to 0, 10, 20, 30, 40, and 50 mg/L to treat the cell wall deficient strain CC-849 and normal cell wall strain CC-124, respectively. The two algal strains were set with the same initial cell density, cultured under the same conditions for three days, and sampled in a 10 mL TAP medium containing different concentrations of inhibitor. Finally, the algal cells affected by the inhibitor were observed microscopically, and the permeability of the cell wall was measured according to Section 4.2.

### 4.8. Transformation of Arabinosyltransferase Gene

#### 4.8.1. Construction of Carrier

The arabinosyltransferase gene (*RRA*) is one of the most crucial function genes related to the synthesis of plant extensin. According to the results of transcriptome sequencing (4.5, 4.6) and enzyme activity inhibition experiment (4.7), arabinosyltransferase may play an important regulatory role in the cell wall synthesis in *C. reinhardtii*. Therefore, the *RRA* gene was knocked out to study its function.

The construction of the *RRA* knockout vector 007S-*RRA*-*Hyg* was based on the pClone007S vector. First, the 1107 bp *RRA* gene was inserted into the vector to obtain the plasmid 007S-*RRA.* Then, the 007S-*RRA* plasmid was digested with restriction endonuclease *Sac*I and *Bsiw*I. Concurrently, T4 ligase was used to link the β-tubulin promoter of *C. reinhardtii*, the *aph7* gene, and the 3’ untranslated region of the *rbcS2* to form the *Hyg* gene expression framework. Finally, by inserting the expression framework of the *Hyg* resistance gene with the homologous arm of the 007S-*RRA* plasmid into the *RRA* gene sequence using a DNA recombinant cloning kit (Vazyme, Nanjing, China), the knockout vector 007S-*RRA*-*Hyg* was obtained. The obtained plasmid 007S-*RRA*-*Hyg* was sequenced to verify that the sequence was correct and that the components in the cassette were connected correctly.

#### 4.8.2. Glass Bead Transformation in *C. reinhardtii*

The transformation process in *C. reinhardtii* is based on the method described in Kindle, which used sterilized glass microbeads to transform [31]. The cells were cultured to a cell density of 3–5 × 10^6^ cells per milliliter and then harvested by centrifugation at 3000× *g* for 10 min. The cell density of 10^8^ cells per milliliter was obtained by re-suspension precipitation using a TAP medium. The cell re-suspension was then swirled for 45 s on a vortex oscillator with 400 μg sterilized glass beads (diameter 425–600 μm) and 3–10 μg circular plasmid DNA. The mixture was inoculated into 5 mL fresh TAP medium, oscillated at 50 rpm at 23 °C, and cultured in a dark environment for 18 h. The algal fluid was then concentrated by harvesting the cells by centrifugation. The 100 μL concentrated *C. reinhardtii* solution was gently and evenly spread on a TAP medium plate containing 1.5% Agar (containing 10 μg/mL hygromycin B). The plates were cultured upside down for 10–15 days, then monoclones were selected from the emerging algal colonies.

#### 4.8.3. Screening of Mutants by Polymerase Chain Reaction

Monoclones were selected from the plate and inoculated into a 5 mL TAP liquid medium containing hygromycin B. The culture oscillated for three days, and then primers were designed according to the arabinosyltransferase gene and hygromycin B gene for PCR (Table 6). According to the bands displayed on 1% agarose gel electrophoresis and gene sequencing, the transgenic algal strains, which were named 124-SR, were verified.

The permeability and integrity of the *C. reinhardtii* cell wall were analyzed according to Section 4.2.

## 5. Conclusions

Transcriptome sequencing technology was used to compare the differential expression genes related to cell wall component synthesis in cell wall deficient strain CC-849 in contrast to the normal cell wall strain CC-124. Similar to the glycosylation of the cell wall glycoprotein in the synthesis pathway of higher plant extensin, there is a metabolic pathway in *C. reinhardtii*, in which the expression of the arabinosyltransferase gene is up-regulated. Further restraining the action pathway of this enzyme by inhibitor, it was found that the cell wall of defective algal strain was not affected, while the permeability of the cell wall in normal algal strain was affected. Finally, utilizing genetic engineering, *RRA* was knocked out from CC-124, and the cell walls of the mutant strains showed unclear boundaries, loose structure, and reduced structure levels. It was speculated that inhibiting arabinosyltransferase would affect the glycosylation modification of cell wall proteins and further affect the assembly of cell wall glycoproteins.

The study of the structure and function of cell wall glycoproteins, as well as the further exploration of the cross-linking and assembly mechanism of cell wall components, has important guiding significance for understanding the evolution of cell wall glycoproteins in algae and higher plants. Based on a deep comprehension of the functions of these genes, gene transformation, gene editing, and other genetic engineering methods will provide new ideas for using these genes to construct more cell wall deficient strains suitable for different purposes such as aquatic feed, health food, drug delivery and so on.

## Figures and Tables

**Figure 1 ijms-24-17595-f001:**
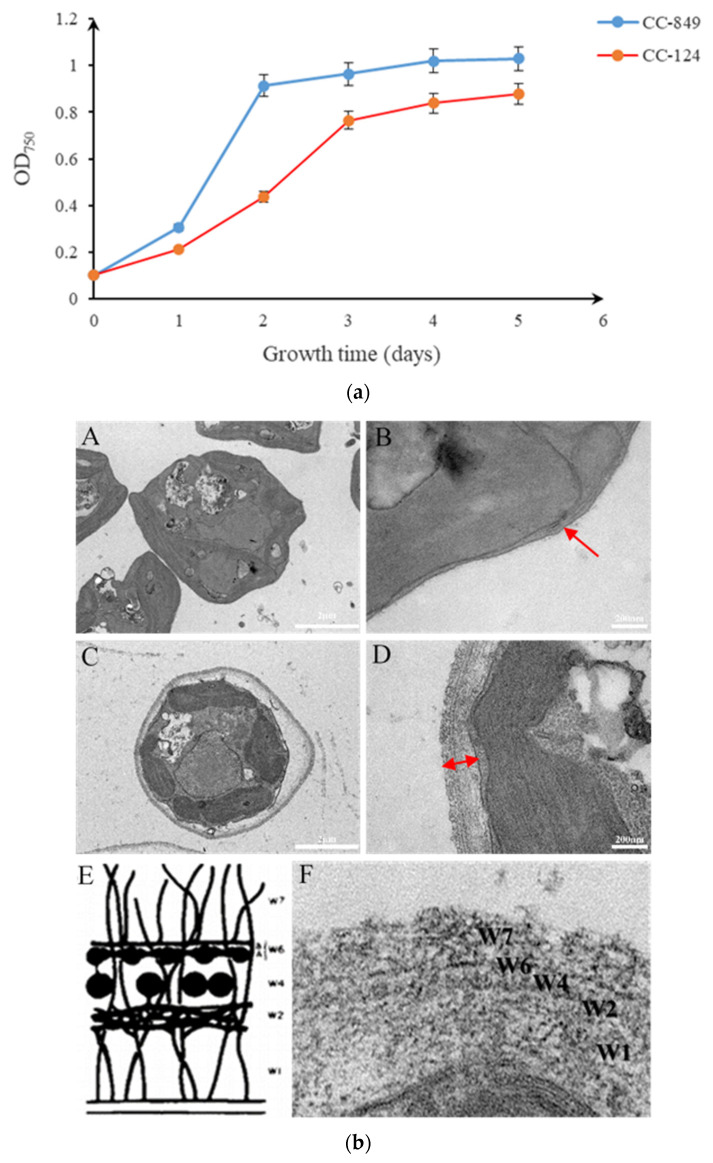
Growth curves and transmission electron micrographs of *C. reinhardtii.* (**a**) The two algal strains were set with the same initial cell density and cultured under the same conditions. The growth trend of the two algal strains in a week with logarithmic and stable phases. (**b**) (**A**,**B**), respectively, show the algal cell morphology of CC-849 and its defective cell wall structure, while (**C**,**D**) show the algal cell morphology of CC-124 and the structural layers of the complete cell wall. The red arrows shown in (**B**,**D**) indicate the difference in the cell wall between CC-849 and CC-124. (**F**) further shows from the outer plasma membrane of the cell, cell wall layers are W1-W7 successively, among which W2, W4, and W6 layers are central glycoprotein lattice structure, which is consistent with the mode diagram (**E**) by Goodenough et al. [9].

**Figure 2 ijms-24-17595-f002:**
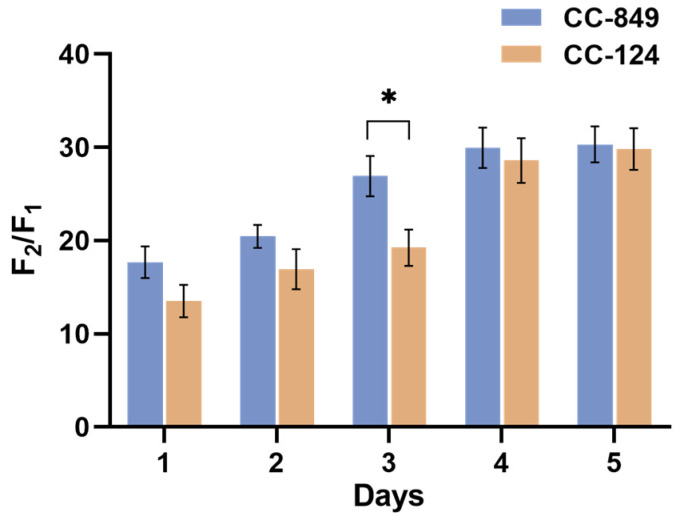
Sensitivity of the two kinds of algal cells to NP-40. Set with the same initial cell density, the two algal strains were sampled every day to detect the sensitivity to NP-40. The figure showed that CC-849 was more sensitive to NP-40 than CC-124 during the growth period, with greater cell wall permeability. * indicates a significant difference between two algal strains on the third day (*p* < 0.05). The ordinate in the figure is the ratio of fluorescence value measured after and before the reaction with NP-40, and the larger fluorescence ratio F_2_/F_1_ indicates the larger cell wall permeability; the abscissa represents the time of sampling.

**Figure 3 ijms-24-17595-f003:**
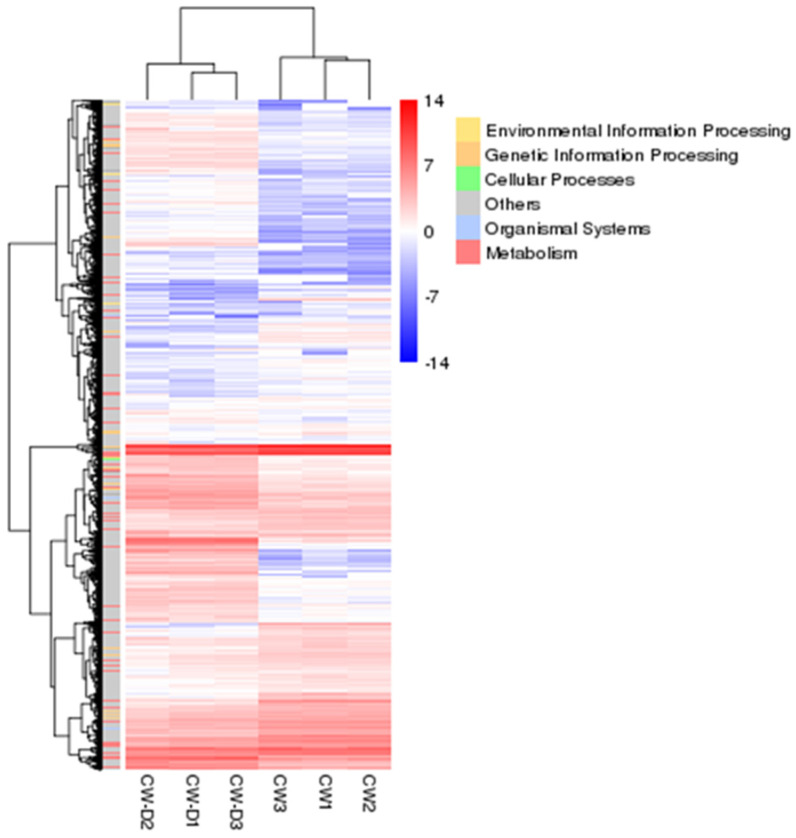
Clustering heat map of differentially expressed genes; the abscissa in the figure is the names of samples and the clustering results of the samples’ genetic relationship, the ordinate is the annotation to KEGG of differential genes, and the clustering results of the predicted genes’ functions, and the color represents the expression level of the genes in the sample. The redder the color, the higher the expression, and the bluer the color, the lower the expression. The legend is the function of gene annotation to the KEGG database.

**Figure 4 ijms-24-17595-f004:**
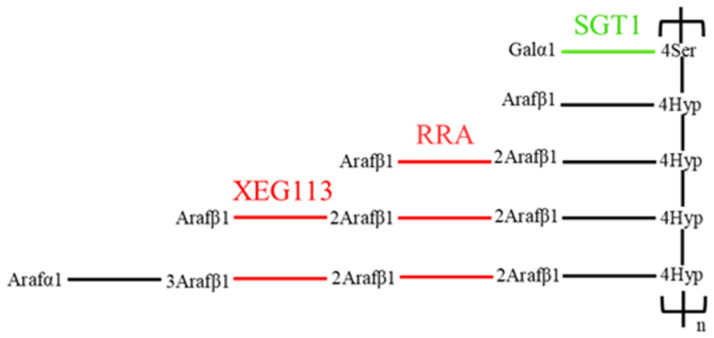
The main differentially expressed genes and related metabolites in the synthetic pathway of extensin in *C. reinhardtii*. Green indicates down-regulation of related genes, and red indicates up-regulation in the comparison group of CC-849 vs. CC-124. *XEG113* (XM_043069322.1), *RRA* (XM_001695456.2), *SGT1* (XM_043068373.1).

**Figure 5 ijms-24-17595-f005:**
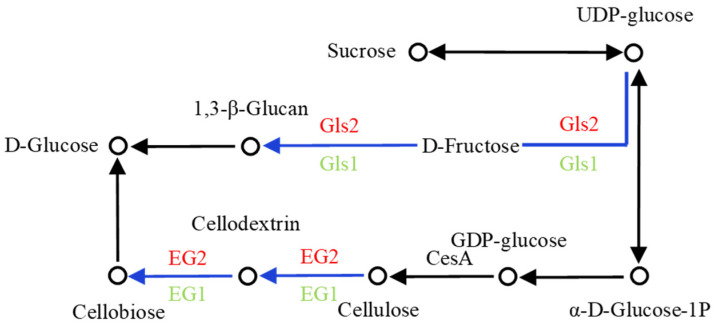
The main differentially expressed enzymes and related metabolites in the synthetic and degraded pathway of glycoskeleton in *C. reinhardtii*. Blue indicates that there are several differentially expressed genes coding one enzyme in the pathway in the comparison group of CC-849 vs. CC-124, with some genes up-regulated and some downregulated.

**Figure 6 ijms-24-17595-f006:**
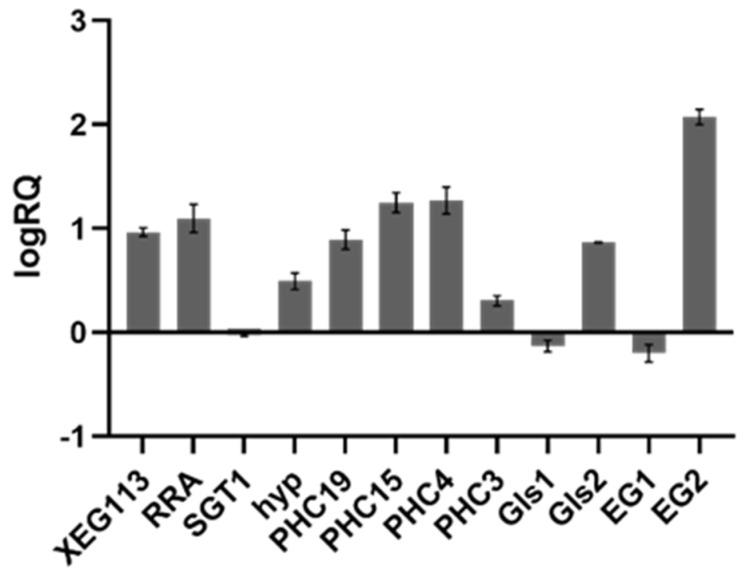
Real−time quantitative PCR verification results. The trend of the gene expression was consistent with the transcriptome sequencing results, proving that the transcriptome results were effective. The ordinate in the figure is the value of logRQ, logRQ > 0 means the gene is up-regulated, and logRQ < 0 means the gene is downregulated. The abscissa is the gene name.

**Figure 7 ijms-24-17595-f007:**
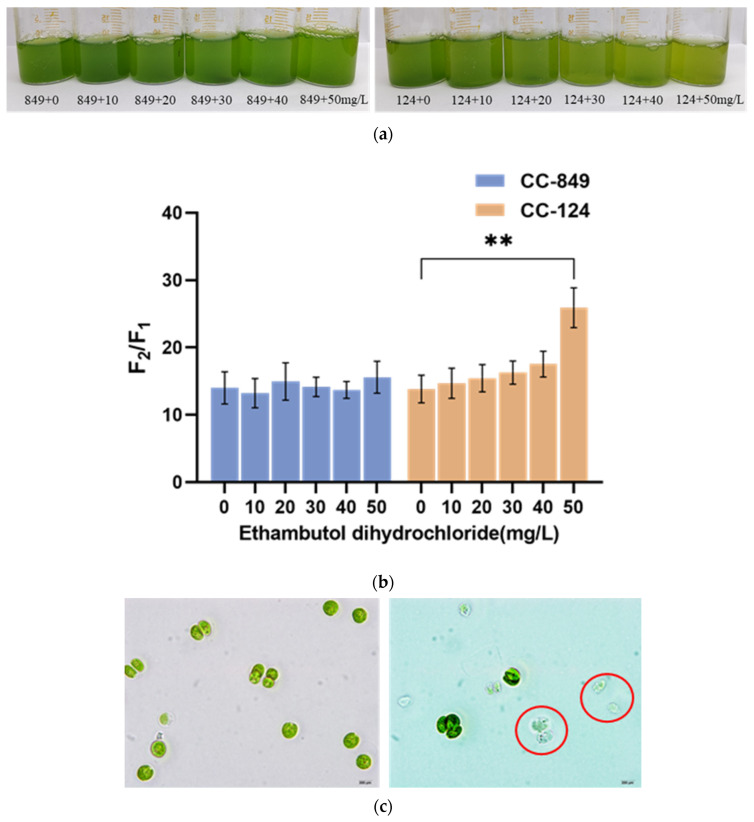
Effects of arabinosyltransferase inhibitor on the two strains of *C. reinhardtii* (**a**) The appearance of two strains treated with an inhibitor of 0, 10, 20, 30, 40, 50 mg/L, the left picture is CC-849, and the right is CC-124. (**b**) The sensitivity of two algae to NP-40 was detected after treatment with different concentrations of the inhibitor. The ordinate in the figure is the ratio of fluorescence value measured before and after the reaction with NP-40; the abscissa sequentially represents CC-849 and CC-124 treated with 0–50 mg/L inhibitor. ** indicates a significant difference between untreated control CC-124 and CC-124 treated with 50 mg/L inhibitor (*p* < 0.01). (**c**) The cell morphology of untreated control CC-124 (**left**) and CC-124 treated with 50 mg/L inhibitor (**right**). The right figure clearly shows more transparent algal cells, as red circles indicate, suggesting that the cell morphology and structure of CC-124 are greatly affected by the 50 mg/L inhibitor.

**Figure 8 ijms-24-17595-f008:**
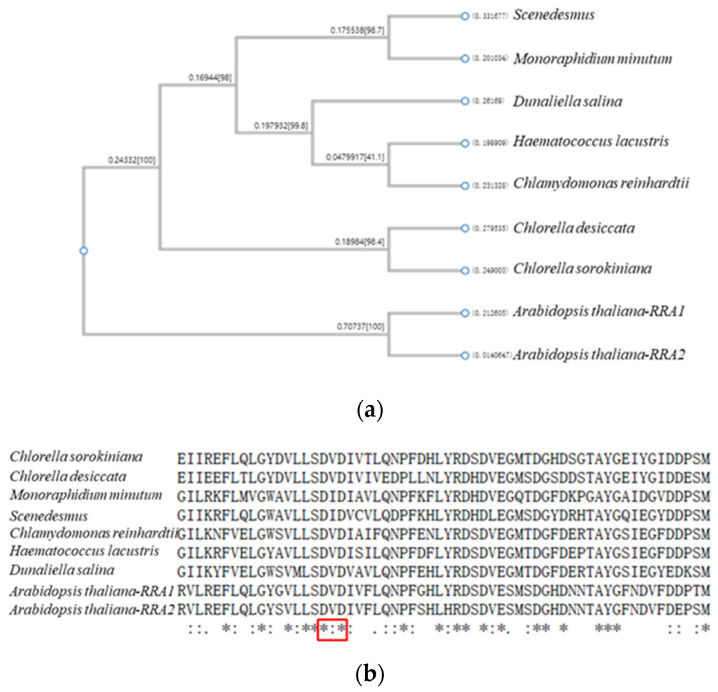
(**a**) is a phylogenetic tree showing genetic relationships of amino acids encoded by the *RRA* gene among algae and *Arabidopsis thaliana* by the Blast protein sequence of *C. reinhardtii*. (**b**) is the homologous amino acid sequence in *C. reinhardtii*, other algae, and *Arabidopsis thaliana*, with the same conserved domain comprising the DxD motif (indicated by a red box). “*” indicates that the sequences involved in the alignment are exactly the same at that site. “:” indicates that conservative substitutions can be observed in this column sequence. “.” indicates that semi-conservative substitutions can be observed in this column sequence.

**Figure 9 ijms-24-17595-f009:**
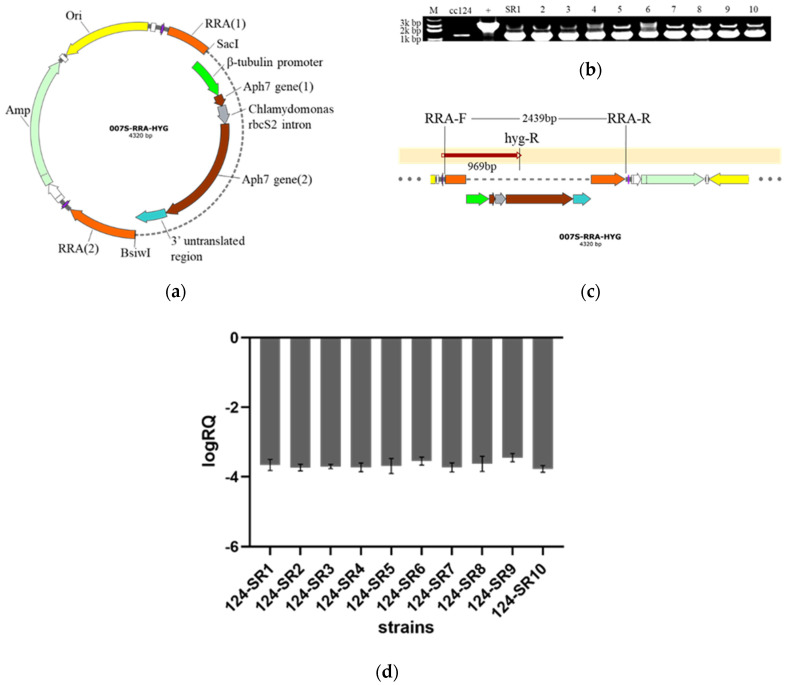
(**a**) is the map of the transformation plasmid, i.e., arabinosyltransferase gene knockout vector 007S−*RRA*−*Hyg*. (**b**) is the PCR identification of positive clones of transformed strains on 1% agarose gel electrophoresis; from left to right is Marker, CC-124, plasmid, 124−SR1−10. The result shows that 124−SR1−10 all have the same bands at 2439 bp with plasmid cloned by RRA−F and RRA−R primers. (**c**) is the sequencing comparison between PCR bands and knockout vector. 969 bp band cloned by RRA-F and hyg−R primers is identical to the fragment of the knockout vector. (**d**) is real-time quantitative PCR results of the transcription level of the *RRA* gene in the transformed strains. The ordinate in the figure is the value of logRQ, and the abscissa is the knocked-out algal strains. LogRQ < 0 means the gene is downregulated, and its lower value shows the larger difference. The expression level of the *RRA* gene was lower in each knocked-out algal strain than that in the control CC-124.

**Figure 10 ijms-24-17595-f010:**
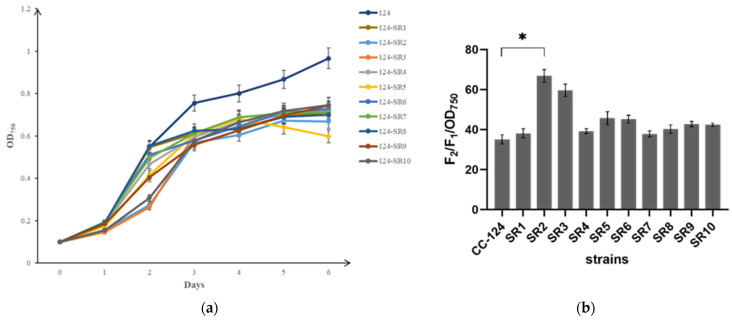
Comparative analysis of cell wall among mutants 124-SR1-10. (**a**)The growth curve of the mutant strains 124-SR1-10. (**b**) The sensitivity to NP-40 of the mutant strains 124-SR1-10. The ordinate in the figure is the ratio of F2/F1, ruling out the effect of cell density OD_750_ to greatly represent the strength of cell wall permeability; the abscissa is the control strain and all mutants. * indicates a significant difference between CC-124 and 124-SR2 strain (*p* < 0.05). (**c**) The surface of the algal cell wall was labeled by FITC-RCA I. Strong green fluorescence indicated algal cell walls with more glycogroup that can bind FITC-RCA I. (**d**) The fluorescence intensity of FITC-RCA I in the single algal cell between the control and the mutants. *** indicates a great significant difference (*p* < 0.001). (**e**) Transmission electron micrographs of CC-124, 124-SR2 and 124-SR8. (**A**–**C**) are electron microscopy images of CC-124, (**A**) is the view of overall algal cells of CC-124, (**B**) is the view of a single algal cell, and (**C**) is used to further observe the structure of the cell wall. Similarly, (**D**–**F**) belongs to 124-SR2, and (**G**–**I**) belongs to 124-SR8.

**Figure 11 ijms-24-17595-f011:**
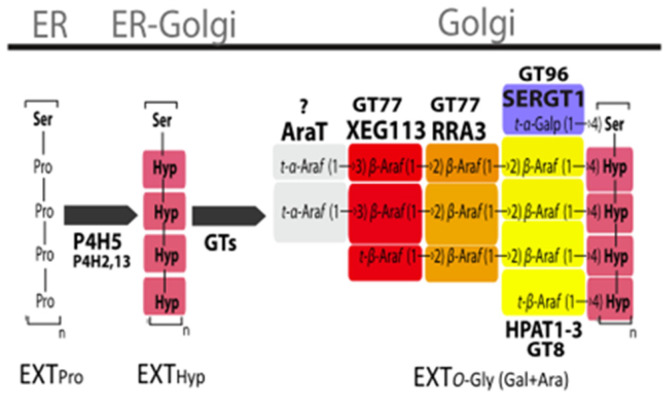
The figure is the post-translational modification steps of EXT and EXT-related proteins encoded by genes cited by Velasquez [25]. Proline hydroxylase genes (*P4Hs*) convert Pro to Hyp. Hyp is then glycosylated by the sequential addition of arabinosyl by arabinosyltransferases encoded by *HPAT1-3*, *RRA3,* and *XEG113* genes. Additionally, Ser is mono-O-galactosylated by peptidyl-Ser galactosyltransferase encoded by the *SERGT1* gene.

**Table 1 ijms-24-17595-t001:** List of data output quality and comparison with the reference genome.

Samples	Clean Reads	Total Reads	GC Content	% ≥ Q30	Mapped Reads	Uniq Mapped Reads
CW-D1	26,528,370	53,056,740	62.32%	95.11%	96.37%	92.79%
CW-D2	24,766,157	49,532,314	62.35%	95.32%	96.42%	92.80%
CW-D3	29,345,506	58,691,012	62.37%	95.10%	95.97%	92.42%
CW1	22,244,162	44,488,324	62.57%	95.29%	96.22%	92.40%
CW2	26,357,107	52,714,214	62.57%	95.48%	95.92%	92.12%
CW3	25,549,348	51,098,696	62.49%	95.43%	95.78%	91.78%

**Table 2 ijms-24-17595-t002:** List of the number of DEG annotations.

Database	Number of DEG	Percentage (%)
Annotated in COG	1323	31.46
Annotated in GO	2537	60.33
Annotated in KEGG	1872	44.52
Annotated in KOG	1381	32.84
Annotated in NR	4201	99.90
Annotated in Pfam	2559	60.86
Annotated in Swiss-Prot	1322	31.44
Annotated in eggNOG	2054	48.85

**Table 3 ijms-24-17595-t003:** List of cell wall structural protein genes in the transcriptome.

Gene Name	NR Annotation Function	Log_2_FC	Regulated
*hyp*	Hydroxyproline-rich cell wall protein	1.601	up
*PHC19*	Cell wall protein pherophorin	5.902	up
*PHC15*	Cell wall protein pherophorin	3.302	up
*PHC4*	Cell wall protein pherophorin	5.939	up
*PHC3*	Cell wall protein pherophorin	3.538	up

**Table 4 ijms-24-17595-t004:** List of measurement data of cell wall thickness of mutant strains.

Algal Strains	The Thickness of the Cell Wall (nm)
CC-124	186.516 ± 18.611
124-SR2	66.465 ± 10.434 ***
124-SR8	168.614 ± 12.081 **

** indicates a significant difference (*p* < 0.05) *** indicates a great significant difference (*p* < 0.001).

**Table 5 ijms-24-17595-t005:** List of qPCR genes and primers.

Gene Name	Primers
*β-tubulin*	F: CGCGTGTCTGAGCAGTT R: CAGGTCGTTCATGTTGGA
*XEG113*	F: TGGGCAAGTCACAATAAGGG R: AACCGCATCCAACAGCATC
*RRA*	F: GCACCAGTGAAAGAAGCATTACC R: GCACTCAGCACCTACCAAACA
*SGT1*	F: CACGGTATTTTCCCTGAGCC R: GAGATGTGGTAATCGCAGAAGG
*hyp*	F: AGTTGGAGGTTCAGCAATGGA R: AAATAGGGAGTCAGGAAGAGGAGT
*PHC19*	F: CAGCAAGGCGGACTTTTACA R: CGGAGCCAAACACCTCAAAT
*PHC15*	F: AAGTGCTGCCCCGTCTACAA R: ACCCAGCAATCACAGCCACA
*PHC4*	F: GAAGTGCCTGGGGCTCATTA R: TTGCTTGCCTGCTGTTTTG
*PHC3*	F: TGACTTGTTTGCGTGGATTGTG R: CGATGCGAAGGGTGTAATGG
*Gls1*	F: TGATTACTGGATGCGACTTGG R: AGCGAGCAATCTTCACGGA
*Gls2*	F: GCTTCCTTCTTTCCCTTCTCA R: TGGTGTTCTGGATGCGGTCT
*EG1*	F: ATGGACCTCGGCACCAAACT R: CACTGCGAAATGCCGAAAC
*EG2*	F: AACAGGACTGAGTAAGAGCGAAGA R: CCAAACGGACTACCAGTGAATGA

**Table 6 ijms-24-17595-t006:** List of PCR genes and primers for screening of mutants.

Gene Name	Primers
*RRA*	F: ATGGGGCTCAAAGACGACTAC R: TTACCGTGTGCCCTTCTCGC
*hyg*	F: CGGCGAGAGCACCAACCCCGTAC R: CTCTCCGGACCGCACCAGTG

## Data Availability

Data are contained within the article and Appendix A.

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
