# Peer review of "Based on Transcriptome Sequencing of Cell Wall Deficient Strain, Research on Arabinosyltransferase Inhibition’s Effect on the Synthesis of Cell Wall in Chlamydomonas reinhardtii"

_ijms, 2023, doi:10.3390/ijms242417595_

Round 1

Reviewer 1 Report

Comments and Suggestions for Authors

Major concerns

1. The time of samples for structural analysis should be further confirmed, since the growth stages and conditions at the first and third day are totally different. It is better to provide some morphological pictures of these stages.

2. Figure 1b should be rephrased. The differences between the two trains should be marked in the picture. The F should be rephrased into high resolution and listed before E.

3. The function and full name of NP-40 should be described when first appearance.

4. What does the y axis mean in Figure 2? Which should be explained in figure legend.

5. The sample names of part 2.2 and Table 1 should be linked to the part 2.1 and fully described when first appearance, together with the sampling time.

6. The details of all genes generated in part 2.3 should be listed in supplementary materials.

7. The KEGG enrichment results should be fully described, which is an important guideline for mining candidate genes. Besides, none of the pathways related to cell wall were enriched in part 2.3.3. How to explain this?

8. The accessions of all genes in part 2.4.1 should be added according to public genome database.

9. A correlation analysis between RNA-seq and qRT-PCR results should be added in part 2.5.

10. Figure 7c should be listed before 7b.

11. The part 2.7.1 and 2.7.2 should be combined, together with Figure 8 and 9.

12. The figure 10 should be rephrased into a beautiful layout.

13. The results of Figure 11b and 11c should be moved to result part and listed at the front of knockout result.

14. The genome version for RNA-seq analysis should be listed in materials and methods.

15. The language should be thoroughly revised.

Comments on the Quality of English Language

A grammar check is necessary.

Reviewer 2 Report

Comments and Suggestions for Authors

Based on transcriptome sequencing of cell wall deficient strain, research of arabinosyltransferase inhibition’s effect on the synthesis of cell wall in Chlamydomonas reinhardtii
The Authors presented a paper regarding the study of the genes involved in the cell wall synthesis in an alga, Chlamydomonas reinhardtii.
The introduction is exhaustive and outlines the scientific sector in which the study fits. The experimental part is well designed, the various steps follow one another in a logical manner and the results are reported and analyzed in a way that is understandable even to those who are less familiar with the topic and the techniques used. In fact, the materials and methods paragraph is extremely detailed and provides useful information to those wishing to carry out similar studies. the English language is good, just needs minor editing. The discussion of the results is in-depth and also examines the most recent literature.
some notes for Authors:
-    check punctuation page 5 at sentence: “According to the statistics of results, there were 4259 differentially expressed genes, among which the number of up-regulated genes 2365 was significantly more than the number of down-regulated genes 1894” better. , 2365 and, 1894 or between brackets
-    golgi is Golgi

Comments on the Quality of English Language

minor editing required

Round 2

Reviewer 1 Report

Comments and Suggestions for Authors

The authors have addressed most of the comments. The manuscript could be accepted after grammar checking.

Comments on the Quality of English Language

A grammar checking is necessary according to the tracked changes.
